# Regulation of Endoplasmic Reticulum–Mitochondria Tethering and Ca^2+^ Fluxes by TDP-43 via GSK3β

**DOI:** 10.3390/ijms222111853

**Published:** 2021-11-01

**Authors:** Caterina Peggion, Maria Lina Massimino, Raphael Severino Bonadio, Federica Lia, Raffaele Lopreiato, Stefano Cagnin, Tito Calì, Alessandro Bertoli

**Affiliations:** 1Department of Biomedical Sciences, University of Padova, 35131 Padova, Italy; federica.lia@phd.unipd.it (F.L.); raffaele.lopreiato@unipd.it (R.L.); tito.cali@unipd.it (T.C.); 2CNR—Neuroscience Institute, 35131 Padova, Italy; marialina.massimino@cnr.it; 3Department of Biology, CRIBI Biotechnology Center, University of Padova, 35131 Padova, Italy; raphaelbonadio@gmail.com (R.S.B.); stefano.cagnin@unipd.it (S.C.); 4CIR-Myo Myology Center, University of Padova, 35131 Padova, Italy; 5Padova Neuroscience Center, University of Padova, 35131 Padova, Italy

**Keywords:** amyotrophic lateral sclerosis, TDP-43, ER–mitochondria contacts, calcium homeostasis, SPLICS, neurodegenerative disorders, GSK3β

## Abstract

Mitochondria–ER contacts (MERCs), tightly regulated by numerous tethering proteins that act as molecular and functional connections between the two organelles, are essential to maintain a variety of cellular functions. Such contacts are often compromised in the early stages of many neurodegenerative disorders, including amyotrophic lateral sclerosis (ALS). TDP-43, a nuclear protein mainly involved in RNA metabolism, has been repeatedly associated with ALS pathogenesis and other neurodegenerative diseases. Although TDP-43 neuropathological mechanisms are still unclear, the accumulation of the protein in cytoplasmic inclusions may underlie a protein loss-of-function effect. Accordingly, we investigated the impact of siRNA-mediated TDP-43 silencing on MERCs and the related cellular parameters in HeLa cells using GFP-based probes for MERCs quantification and aequorin-based probes for local Ca^2+^ measurements, combined with targeted protein and mRNA profiling. Our results demonstrated that TDP-43 down-regulation decreases MERCs density, thereby remarkably reducing mitochondria Ca^2+^ uptake after ER Ca^2+^ release. Thorough mRNA and protein analyses did not highlight altered expression of proteins involved in MERCs assembly or Ca^2+^-mediated ER–mitochondria cross-talk, nor alterations of mitochondrial density and morphology were observed by confocal microscopy. Further mechanistic inspections, however, suggested that the observed cellular alterations are correlated to increased expression/activity of GSK3β, previously associated with MERCs disruption.

## 1. Introduction

It is now widely accepted that organelles within cells are strictly connected with each other through the transmission of soluble signalling molecules and the formation of dynamic platforms of inter-organellar connections called membrane contact sites (MCSs). MCSs maintain adjacent membranes in close proximity thanks to the presence of several molecules/proteins acting as molecular bridges and crucially contribute to the functional cross-talk between organelles [1,2].

Among different MCSs identified within cells, the most studied is the extensive network of structures between mitochondria and the endoplasmic reticulum (ER), also referred to as mitochondria-associated membranes or mitochondria–ER contacts (MERCs) [3,4,5,6,7]. Through MERCs, mitochondria and the ER lumen directly and continuously communicate with each other regulating key physiological processes [5], such as mitochondria biogenesis [8] and dynamics [9,10], autophagosome formation [11] and lipid synthesis and transfer [12,13,14].

One of the major players in ER–mitochondria communication is Ca^2+^ [15,16,17,18,19]. Ca^2+^ exchange between ER and mitochondria needs to be tightly regulated because it is essential for key mitochondrial functions [16,17,19], e.g., the regulation of key enzymes of the tricarboxylic acid cycle and cellular Ca^2+^ buffering [20,21], while excessive mitochondrial Ca^2+^ accumulation can lead to cell death by triggering the opening of the mitochondrial permeability transition pore [22].

It is fair to acknowledge that once such finely tuned ER–mitochondria interplay is perturbed, several cell functions fail and pathogenic mechanisms become activated. This is the case for different neurodegenerative diseases [18,23], also including amyotrophic lateral sclerosis (ALS) [23,24,25,26,27]. Indeed, MERCs disruption, possibly associated with deregulated Ca^2+^ homeostasis [27,28,29,30], mitochondrial abnormal morphology and biochemical dysfunctions [31,32,33], was previously reported as an ALS hallmark.

Currently, no treatment has been identified that may efficiently halt or cure ALS pathology, a devastating neurodegenerative disease characterised by the progressive loss of motor neurons, resulting in paralysis, respiratory failure and ultimately death within 2–5 years from diagnosis [34]. The identification of reliable therapeutic targets was hampered so far by the wide heterogeneity of genetic, clinical and biochemical features of ALS [35]. An important breakthrough into ALS pathogenesis has been offered by the identification of the (TAR) DNA binding protein 43 (TDP-43) as the major component of cytoplasmic inclusions found in the majority of ALS cases (up to 97%) [36,37,38]. In addition, 3–5% of familial ALS cases (covering 5–10% of total ALS cases) results from inherited mutations in the TDP-43-encoding gene [39,40]. To date, however, disease mechanisms triggered by TDP-43 have not been clearly elucidated.

TDP-43 is a highly conserved member of the heterogeneous ribonuclear protein family, characterised by a ubiquitous expression. Mainly localised in the nucleus, TDP-43 shuttles constantly between nucleus and cytoplasm in a transcription-dependent manner [41]. In both compartments, TDP-43 is involved in several steps of RNA processing, including transcription, translation, alternative splicing, mRNA transport and stability, and in microRNA and long non-coding RNA processing [38,42,43,44,45]. TDP-43 was reported to associate to thousands of mRNA targets [46,47,48,49] and bind with high specificity to UG-rich sequences in selected 3′ UTRs of mRNAs/pre-mRNAs, thus regulating mRNA stability and fate [50].

In keeping with these fundamental functions, cytoplasmic TDP-43 mislocalisation and aggregation in ALS-affected neurons are supposed to trigger neurodegeneration not only by the acquisition of a potentially toxic activity in the cytoplasm (“gain-of-function” hypothesis) but also by the loss of the protein’s physiological functions (“loss-of-function” hypothesis) [51,52,53,54].

In the attempt to clarify whether TDP-43 loss of function disrupts ER–mitochondria cross-talk as a possible pathogenic mechanism in TDP-43 proteinopathies, in the present study, we analysed MERCs and ER–mitochondria Ca^2+^ fluxes by different genetic approaches in HeLa cells. To this purpose, we down-regulated TDP-43 expression by an already validated targeted siRNA [55] and quantitatively analysed MERCs presence and ER–mitochondria Ca^2+^ signalling using novel split green fluorescent protein-based contact site sensors (SPLICS) [56,57,58] and genetically targeted aequorin Ca^2+^ probes, respectively, in comparison with control HeLa cells. The reported results suggest that TDP-43 participates in the maintenance of proper ER–mitochondrial signalling, possibly through the regulation of RNA metabolism and glycogen synthase kinase 3β (GSK3β) expression/activity.

## 2. Results

### 2.1. TDP-43 down-Regulation Affects ER–Mitochondria Tethering

The loss of TDP-43 functionality caused by its mislocalisation and accumulation into cytoplasmic inclusions (found in over 95% of ALS cases) may contribute to processes governing ALS pathogenesis. One of these is represented by the failure of ER–mitochondria communication due to the disruption of MERCs [29,59,60,61], although the underlying mechanisms are not yet well-defined.

In light of the above considerations, here, we investigated whether the down-regulation of TDP-43 causes alterations of MERCs in HeLa cells using a siRNA-based approach for TDP-43 expression silencing. Despite the considerable down-regulation of TDP-43 in HeLa cells transfected with TDP-43 siRNA oligonucleotides (by approximately 70%), with respect to control cells transfected with scrambled siRNA oligonucleotides (Figure 1a), no alteration in cell viability was observed in TDP-43 knock-down cells (Figure 1b), thus indicating that acute TDP-43 depletion does not dramatically influence cell homeostasis. To explore the impact of TDP-43 knock-down on ER–mitochondria proximity, we transfected HeLa cells with plasmids coding for two different split-GFP-based contact site sensor (SPLICS) probes useful to follow either short-range (SPLICS_S_) (Figure 1c) or long-range (SPLICS_L_) (Figure 1d) ER–mitochondria interactions and engineered to fluoresce exactly at the ER–mitochondria interface [58,62]. Surprisingly, the loss of TDP-43, which is predominantly localised in cell nuclei in control cells (as shown in Figure 1c,d, middle panels), drastically reduced the number of both short-range and long-range MERCs per cell.

To ascertain if the toxicity caused by TDP-43 cytoplasmic foci—often occurring under pathological conditions—alters the physical interaction between ER and mitochondria, we also monitored ER–mitochondria networks in HeLa cells transfected with either wild-type (WT) TDP-43 or the ALS-related Q331K TDP-43 mutant. Surprisingly, the number of ER–mitochondria contacts was not perturbed by the overexpression of (WT or mutated) TDP-43 (Appendix A), which we recently demonstrated to cause apoptotic cell death in another human cell line [63].

We can thus conclude that, despite the presence of TDP-43-positive cytoplasmic inclusions (arrows in Appendix A), TDP-43 (either WT or mutated) overexpression does not impinge on ER–mitochondria tethering. Conversely, functional TDP-43 is involved in maintaining a correct ER–mitochondria cross-talk.

### 2.2. Dysregulation of Mitochondria Ca^2+^ Uptake and ER Ca^2+^ Discharge in TDP-43 Knock-Down HeLa Cells

A proper ER–mitochondria cross-talk is essential to regulate Ca^2+^ homeostasis, whose alteration represents another hallmark of ALS and of other neurodegenerative diseases.

To evaluate if TDP-43 down-regulation impairs the functional counterpart of ER–mitochondria physical interaction, we measured ER–mitochondria Ca^2+^ transfer using genetically encoded aequorin (AEQ) probes. In particular, we employed the mitAEQ probe targeted to the mitochondrial matrix. This probe was used to monitor ER–mitochondria Ca^2+^ shuttling upon treatment with histamine. Histamine is a well-known inositol 1,4,5-trisphosphate (IP_3_)-generating agonist that induces Ca^2+^ release from the ER stores via IP_3_ receptors (IP_3_Rs), with the consequent uptake of the ion by mitochondria through the mitochondrial calcium uniporter (MCU) complex. Figure 2a reports Ca^2+^ transients in the mitochondrial matrix following histamine stimulation in HeLa cells pre-treated with either TDP-43 siRNA or scrambled siRNA oligonucleotides, used as control. These experiments clearly showed that, compared to the control, TDP-43 silencing caused significantly reduced Ca^2+^ transients in the mitochondrial matrix.

The finding of lower mitochondrial Ca^2+^ uptake in TDP-43 knock-down cells prompted us to investigate whether this was due to a lower ER Ca^2+^ buffering capacity or an altered ER Ca^2+^ discharge. This task was accomplished using the AEQ probe targeted to the ER lumen (erAEQ). Briefly, to monitor the ER [Ca^2+^] steady state, erAEQ-expressing HeLa cells were firstly fully depleted of Ca^2+^ and subsequently perfused with 1 mM Ca^2+^ allowing cells to rapidly accumulate the ion in the ER by the sarco-endoplasmic reticulum Ca^2+^ ATPase (SERCA) activity. As reported in Figure 2b, we found no significant alteration in the ER [Ca^2+^] steady state in TDP-43-silenced cells with respect to the non-silenced control. However, monitoring the histamine-induced ER Ca^2+^ discharge via IP_3_Rs, by use of the same erAEQ probe, showed a significantly slower (by approximately 50%) rate of Ca^2+^ release from the ER in TDP-43 knock-down HeLa cells (Figure 2c). We also measured cytosolic Ca^2+^ transients in response to metabotropic stimulation with histamine, showing that they were unaffected by TDP-43 down-regulation (Appendix A).

Considering the above results, we checked whether TDP-43 silencing altered the abundance of ER Ca^2+^-binding proteins, such as calreticulin (CRT) and calnexin (CLNX), and of the unfolded protein response regulator glucose-regulated protein 78 (GRP78), and/or some major Ca^2+^-transporting systems, i.e., the sarco-endoplasmic reticulum Ca^2+^ ATPase (SERCA), and the main Ca^2+^-release channels in the ER, inositol 1,4,5-trisphosphate receptor (IP3R). Western blot (WB) analyses, however, resulted in no significant alteration of the abundance of such proteins in HeLa cells transfected with TDP-43 siRNA or scrambled siRNA oligonucleotides (Figure 3), thus indicating that TDP-43 down-regulation does not perturb the expression of such proteins. Notably, the results for GRP78 and IP3R were confirmed also by the analysis of the corresponding transcripts, showing no alteration in their abundance in the two cell populations (see Section 2.4).

### 2.3. Evaluation of Mitochondrial Parameters in TDP-43 Knock-Down HeLa Cells

Starting from the above-reported results, we scrutinised if TDP-43 silencing might impair mitochondrial abundance/function through the immunodetection of the translocase of the outer mitochondrial membrane 20 (TOM20) protein, a central component of the TOM receptor complex involved in the recognition and translocation of cytosolically synthesised preproteins into mitochondria [64], or by examining the levels of proteins belonging to the mitochondrial respiratory complexes. As shown in Figure 4a and Appendix A, treatment of HeLa cells with TDP-43 siRNA does not induce any alteration of such proteins. We also analysed mitochondria morphology by confocal microscopy of fixed cells stained with an antibody to TOM20, suited for visualising the mitochondrial network (Figure 4b). Different markers of mitochondrial elongation, such as circularity (Figure 4c), mitochondrial skeletal length (Appendix A) and aspect ratio (Appendix A), again revealed no difference in the two populations of HeLa cells transfected with TDP-43 siRNA or scrambled siRNA oligonucleotides. Interestingly, although previous reports suggested a possible mitochondrial localisation for TDP-43 [65], no co-localisation was found between TOM20 and TDP-43 staining, thus supporting the thesis that, under our experimental conditions, TDP-43 does not localise to mitochondria. In addition, gene expression and protein level of the ion-translocating subunit of the MCU complex and of the two regulatory subunits for mitochondrial Ca^2+^ uptake (MICU) 1 and MICU2 did not significantly differ between the two HeLa cell populations (Figure 4d,e). It is worth mentioning, however, that MCU activity is also regulated by other ancillary proteins (i.e., MICU3 and the essential MCU regulator [66]), thus demanding further studies to elucidate their abundance. Parenthetically, the MICU2 transcript level seems to be lower than those of MICU1 and MCU (both present at the same amount); however, we did not perform specific statistical analyses nor scrutinise in more detail such an issue because it is far away from the scopes of the present study.

### 2.4. TDP-43 Alters the Abundance of the ER–Mitochondria Contacts Regulator GSK3β

Given that TDP-43 regulates mRNA biogenesis and maturation, we investigated whether TDP-43 was influencing the expression of specific genes and proteins implicated in ER–mitochondria tethering. To address this question, we first monitored the expression of several of such genes and then tested some of them at the protein level. Specifically, we tested the gene expression of mitofusin-1 and 2 (MFN1 and MFN2), heat shock protein family A (Hsp70) member 9 (HSPA9 or GRP75), voltage-dependent anion channel 1 (VDAC1), inositol 1,4,5-trisphosphate receptor type 3 (IP3R), VAMP associated protein B (VAPB), a regulator of microtubule dynamics 3 (RMDN3 or PTPIP51) and heat shock protein family A (Hsp70) member 5 (HSPA5 or GRP78). In this context, it is worth noting that MFN1 and MFN2 are two key components of MERCs [67], and—together with GRP75, VDAC and the ER-residing IP_3_R—they form one of the most important tethering complexes in MERCs formation and function [68].

None of the analysed transcript, however, was modified by TDP-43 silencing (Figure 5a), and protein analysis of a subset of such genes confirmed the unaltered expression of GRP78 and IP3R (Figure 3), MFN1, MFN2 and GRP75 (Figure 5b).

Previous studies suggested that VAPB and PTPIP51—whose transcripts were not altered by TDP-43 silencing (Figure 5a)—interact to form one of such ER–mitochondria tethering complexes, and the disruption of such interactions have been linked to the pathogenic process of ALS [27,29,59]. The loss of VAPB/PTPIP51 interaction occurs through increased activation of GSK3β, as already demonstrated both in TDP-43 or fused in sarcoma (FUS) models, although the mechanism is still obscure [29,59]. We thus reasoned that TDP-43 silencing in HeLa cells might be responsible for an alteration of GSK3β activity and examined GSK3β expression level and phosphorylation state by transcriptional analysis and WB. We observed that TDP-43 silencing increased the level of GSK3B gene transcript while leaving that of GSK3A unaltered (Figure 5c). In addition, TDP-43 silencing increased the abundance of GSK3β protein while concurrently reducing the inhibitory phosphorylation of GSK3β on Ser9 (Figure 5c). Such findings suggest that TDP-43 loss of function perturbs ER–mitochondria contacts and Ca^2+^ exchange by enhancing GSK3β functionality.

## 3. Discussion

In this work, we provided evidence that substantial siRNA-based down-regulation of the ALS-related protein TDP-43 in HeLa cells significantly disrupted ER–mitochondria tethering and ER–mitochondria Ca^2+^ cross-talk by reducing the number/density of MERCs per cell.

MERCs are specific regions in which mitochondria and the ER are strictly coupled, allowing the reciprocal exchange of biochemical signals and coordinating several cellular functions, such as mitochondrial trafficking and bioenergetics, Ca^2+^ homeostasis and signalling, and cell death [6,69]. Our results are relevant in the field of neurophysiology and neuropathology because MERCs are fundamental in maintaining proper neuronal functions, and their early impairment in neurons has been causally related to aging and neurodegenerative diseases, including ALS [61].

A possible connection between TDP-43 and altered MERC structure and function was already explored, albeit providing conflicting results [29,70]. For example, the overexpression of TDP-43 (either WT or bearing familial ALS-associated mutations) reduces MERCs, leading to Ca^2+^ homeostasis alterations [29]. On the other hand, however, another research reported no alteration in ER–mitochondria tethering upon the overexpression of WT or mutated TDP-43, although under the same conditions, mitochondrial Ca^2+^ uptake was perturbed [70]. These previous studies mainly addressed a TDP-43 toxic gain-of-function, which we agree is a possible major contributor to ALS pathogenesis. The conflicting results from such studies might be due to the different methodological approaches used to analyse ER–mitochondria association, the levels of TDP-43 overexpression and/or the different used cell models.

Here, by using a reverse approach, mainly based on TDP-43 silencing in HeLa cells and other genetic tools, including SPLICS probes for MERCs quantification [56,57,58,62], we demonstrated that TDP-43 down-regulation potently diminished both short- (~8–10 nm) and long-range (~40–50 nm) ER–mitochondria interactions. Importantly, by using the same SPLICS approach, we observed that MERCs density was not significantly perturbed by overexpressing either WT or ALS-related Q331K mutant TDP-43, which led to TDP-43 accumulation in cytotoxic inclusion bodies [63], thereby further suggesting a loss-of-function contribution of TDP-43 in MERCs disruption. This is in line with several previous reports that the loss of TDP-43 functionality contributes to ALS pathogenesis, as suggested by the profound transcriptome-wide changes [71,72] and strong developmental and post-developmental phenotypes [73,74], including neurodegenerative traits [54,75,76,77], caused by TDP-43 knock down.

The next question to answer is how TDP-43 may impinge on MERCs.

It is now well established that TDP-43, having a prevalent nuclear localisation under physiological conditions, may shutter between the nucleus and the cytoplasm and play key roles in RNA biogenesis, processing and transport [42,49]. A mitochondrial localisation of TDP-43 has also been reported [65,78] and related to either pathological outcome [79,80] or physiological functions, such as the maintenance of appropriate levels of specific mitochondrial RNAs and correct mitochondrial function [81]. In our experiments, however, we never observed the presence of TDP-43 in mitochondria, thus ruling out the possibility that mitochondrial TDP-43 depletion by siRNA per se contributes to MERCs destabilisation.

The disruption of ER–mitochondria signalling seems to be a causal event in ALS, and some mechanistic hypotheses have been forwarded for such a correlation. For example, mutations in the Sigma 1 receptor (*SIGMAR1*) related to frontotemporal lobar degeneration (FTLD), co-occurring with ALS (FTLD-ALS) [82] and juvenile ALS [83], cause the disruption of MERCs followed by perturbation of Ca^2+^ homeostasis, activation of ER stress and ultimately motor neuron degeneration [84]. SIGMAR1 is a chaperone protein located at MERCs that regulates the compartmentalisation and export of lipids as well as ER–mitochondria Ca^2+^ fluxes [85].

Further evidence was provided by FUS, a protein involved in DNA repair, transcription and splicing [86], whose accumulation is a hallmark of FTLD-ALS and whose mutations were associated with familial forms of ALS and FTD [87]. Interestingly, disruption of MERCs was also found in FTLD-ALS-related FUS transgenic rodents and in NSC-34 cells overexpressing FUS (either WT or carrying ALS-related mutations) [59]. In this case, the disruption of ER–mitochondria association also occurs through the reduced affinity between VAPB and PTPIP51 [29,59]. Notably, VAPB has been identified to be causative of some dominantly inherited familial forms of ALS type-8, associated with ER stress, alteration of cytoskeleton organisation, ER-to-Golgi transport, bouton formation at the neuromuscular junction and altered Ca^2+^ homeostasis [88,89,90].

To obtain insights into the mechanisms of MERCs disruption in our TDP-43 knock down model, we monitored the expression levels of proteins known to participate in MERCs formation, but we excluded that TDP-43 directly regulates their expression.

In previous works, different proteomics approaches were applied to identify the molecular components/regulators of MERCs at a large-scale level [91,92], and recently, more than one hundred proteins were associated with MERCs formation under physiological conditions [93]. Further studies are thus needed to elucidate if TDP-43 participates in regulating the expression of such proteins. Since, in the present study, mRNA analysis demonstrated altered gene expression of the multifunctional serine/threonine-protein kinase GSK3β upon TDP-43 silencing, we evaluated, using a protein targeted approach, if TDP-43 depletion modified GSK3β. This choice stemmed from the notion that GSK3β activation or inhibition reduces or increases VAPB-PTPIP51 interaction, respectively [59], although the molecular mechanism of such regulation is still unclear. Importantly, the perturbation of GSK3β activity was also implicated in the pathogenesis of ALS and many other neurodegenerative disorders [94,95,96]. For the first time to our knowledge, here, we demonstrated that TDP-43 down-regulation induces both an increased expression and a higher activation (i.e., reduced phosphorylation in Ser9) of GSK3β, allowing us to speculate that this pathway may significantly contribute to the reported reduction in ER–mitochondria tethering after TDP-43 silencing.

Our results also agree with previous data demonstrating that GSK3β expression/activity was increased in post-mortem samples of ALS patients [97,98]. Moreover, signalling pathways regulating GSK3β activation were altered in ALS mouse models [99], while the loss of the GSK3β orthologue in *Drosophila* (*Shaggy*) suppressed TDP-43-induced motor axons and neuromuscular junctions degeneration [100]. Taken together, these data suggest that GSK3β activity is increased in ALS and that its inhibition might rescue defective phenotypes, such as MERCs disruption.

Future investigations are needed to understand the mechanisms underlying the correlation between TDP-43 and GSK3β gene expression and protein level/activity. Considering that one of the major functions of TDP-43 is the regulation of RNA metabolism, one may simply hypothesise a direct post-transcriptional regulation of GSK3β mRNA by TDP-43 or, indirectly, a regulation of microRNAs targeting GSK3β mRNA. In this framework, a very speculative possibility is that an intermediate between TDP-43 and GSK3β is Traf2-and-NcK interacting kinase (TNIK). This hypothesis is supported by previous reports showing that TDP-43 regulated the splicing and the expression level of TNIK [101] and that TNIK ablation (in TNIK^−/−^ mice) was related to increased GSK3β level [102].

As for the possible pathogenic implications of MERCs impairment by TDP-43 knock down, it is of utmost importance to recall the deleterious consequence of defective ER–mitochondria tethering on the correct ER–mitochondria Ca^2+^ cross-talk. Indeed, we here showed that TDP-43 down-regulation causes lower mitochondrial Ca^2+^ uptake upon stimulating the release of the ion from the ER lumen with histamine. Reduced ER–mitochondria Ca^2+^ transfer may impinge on both mitochondrial functions and the overall control of cell Ca^2+^ homeostasis since Ca^2+^ is involved in the control of mitochondrial bioenergetics, and mitochondria act as one of the major Ca^2+^-buffering systems in the cells, avoiding harmful cytosolic Ca^2+^ overloads.

Although we observed that decreased mitochondrial Ca^2+^ uptake capability in TDP-43 down-regulated cells did not induce a concomitant rise in cytosolic [Ca^2+^], which have been often associated with the activation of (apoptotic) cell death pathways, our findings may suggest that TDP-43 loss of function could compromise mitochondrial functionality.

## 4. Materials and Methods

### 4.1. Plasmids

Plasmids SPLICS_S_-P2A^ER–MIT^ and SPLICS_L_-P2A^ER–MIT^, coding for SPLIC_S_ and SPLIC_L_ probes, respectively, and plasmids coding for the Ca^2+^ probe aequorin (AEQ) linked to sequences addressing the protein to different cell compartments (i.e., the bulk cytosol (AEQ_cyt_), the mitochondrial matrix (AEQ_mit_) or the ER lumen (AEQ_ER_) were already described in our previous works [58,62,103,104,105]. The constructs codifying for mKate2 and TDP-43-mKate2 (encoding for both the WT and the ALS-related mutant Q331K) were already available in our laboratory [63].

### 4.2. Cell Cultures, siRNA-Based TDP-43 Silencing and Cell Viability Assay

HeLa cells were grown in Dulbecco’s modified Eagle medium supplemented with 10% (*v*/*v*) foetal bovine serum, 2 mM L-glutamine, 100 U/mL penicillin and 100 μg/mL streptomycin and maintained at 37 °C in a humidified incubator with a 5% CO_2_ atmosphere. For immunofluorescence analyses and Ca^2+^ measurement experiments or for biochemical and RT-PCR assays, cells were seeded at a density of 4 × 10^4^ (onto 13-mm coverslips) or 7 × 10^4^ (onto 12-well plates), respectively. RNA interference was performed according to Romano et al., 2020 [55], with minor modifications. Briefly, immediately after seeding, cells were transfected with 200 nM of either the targeting (TDP-43: 5′-gcaaagccaagaugagccu-3) or scrambled (MISSION^®^ siRNA Universal Negative Control #.1, Sigma-Aldrich, St. Louis, MO, USA) siRNA oligonucleotides. For this purpose, HiPerfect Transfection Reagent (cat. no. 301705, Qiagen, Hilden, Germany) and Opti-MEM I reduced serum medium (cat. no. 51985-026, Thermo Fisher Scientific, Waltham, MA, USA) were used, according to the manufacturer’s instructions. Cells were incubated in the presence of siRNAs for 72 h before subsequent analyses.

Cell viability of HeLa cells transfected with scrambled or TDP-43 siRNA oligonucleotides was determined using the MTS [3-(4,5-dimethylthiazol-2-yl)-5-(3-carboxymethoxy-phenyl)-2-(4-sulfo phenyl)-2H-tetrazolium, inner salt] assay (CellTiter96 Aqueous One 5 Solution Assay, Promega, Madison, WI, USA) following the manufacturer’s instructions, as previously described [63]. Briefly, 72 h after transfection, the cell culture medium was removed from, and the MTS reagent was added to, the 96-well cell culture plates. After incubation (90 min) at 37 °C, the absorbance of reduced MTS (λ = 490 nm) was determined using a Microplate Reader (TECAN, Männedorf, Switzerland) spectrophotometer. Obtained data were then normalised to the mean value of HeLa cells transfected with scrambled siRNA.

### 4.3. Transfection with Plasmids Coding for SPLIC Probes

To quantitatively evaluate MERCs, 24 h after siRNA treatment, cells were transfected with plasmids coding for SPLIC_S_ or SPICS_L_ encoding cDNAs, while in experiments involving TDP-43 overexpression, cells were co-transfected with plasmids coding for TDP-43 (WT) or TDP-43(Q331K) and for SPLIC_S_ or SPLIC_L_ (in a ratio of 1.5:2) 24 h after plating. In all cases, Lipofectamine 3000 transfection kit (cat no. L3000-015, Thermo Fisher Scientific, Waltham, MA, USA) was used, following the manufacturer’s instructions. 48 h after transfection, coverslips were rinsed twice with phosphate-buffered saline (PBS; 140 mM NaCl, 2 mM KCl, 1.5 mM KH_2_PO_4_, 8 mM Na_2_HPO_4_, pH 7.4) and fixed with formaldehyde [3,7% (*v*/*v*) (cat. no. F8775, Sigma Aldrich, St. Louis, MO, USA) in PBS] (20 min, 4 °C). TDP-43 knock-down cells were then processed as described below for immunocytochemistry, while TDP-43 overexpressing cells were mounted onto microscope slides using a commercial fluorescence mounting medium (DAKO, Santa Clara, CA, USA) and observed by confocal microscopy, as described below.

### 4.4. Immunocytochemistry

Fixed cells were permeabilised by incubating cell-containing coverslips in Triton-X100 (0.1% (*w*/*v*) in PBS) (15 min, 4 °C). Cells were then incubated (overnight, 4 °C) with the desired primary antibody (Ab) diluted (as indicated in parentheses) in PBS added with 1% (*w*/*v*) bovine serum albumin (BSA). In detail, we used mouse monoclonal (m) anti-TDP-43 Ab ((1:25) cat. no.SC-376532, Santa Cruz Biotechnologies, Dallas, TX, USA) and anti-Tom20 rabbit polyclonal (p) Ab (1:50, Santa Cruz Biotechnologies, cat. no. sc-11415). After extensive washings in PBS, cells were incubated (1 h, RT) with the following secondary Abs: Alexa Fluor 594-conjugated anti-mouse IgG (1:500, cat.no. A-21203, Molecular Probes, Thermo Fisher Scientific, Waltham, MA, USA) and Alexa Fluor 488-conjugated anti-rabbit IgG (1:200, cat.no. A-11034, Molecular Probes, Thermo Fisher Scientific, Waltham, MA, USA). Cell nuclei were counterstained with Hoechst 33,342 (5 μg/mL, Sigma-Aldrich St. Louis, MO, USA), and after further washings in PBS, coverslips were mounted as described above and observed by confocal microscopy.

### 4.5. Confocal Microscopy, Image Acquisition and Processing

For the evaluation of either ER–mitochondria contact sites or mitochondrial morphology, cells were observed with a Leica SP5-TCS-II-RS inverted confocal microscope using laser illumination at 405, 488 and 594 nm wavelengths depending on the fluorescent probes using an HCX Plan APO 63X (numerical aperture, 1.40) oil-immersion objective. For all images, the pinhole was set to 1 Airy Unit. Confocal microscopy imaging was performed at 1024 × 1024 pixels per image, with a 0.2 Hz acquisition rate. To avoid cross-talk between fluorophores, sequential scans (using the option “between frames”) were performed. Then, a Z-stack was acquired through the whole thickness of the cell at 290 nm intervals in the z-plane. Cell images were elaborated using the Fiji software [106] and were imaged in the figures as the merge of total Z-stacks. To quantify ER–mitochondria contacts sites, a complete Z-stack was processed using the ImageJ software (National Institutes of Health, Bethesda, Maryland, USA) as previously described [62]. For this analysis, at least 6 different fields from 3 independent biological replicates (i.e., different transfections) were used. To evaluate mitochondrial morphology, 60 *z*-axis images collected at 0.2 μm intervals were analysed using the Volocity 6.0.0 3D Image Analysis Software (Perkin Elmer, Waltham, MA, USA, RRID:SCR_002668). Each identified mitochondrion was analysed for its morphological characteristics, such as skeletal length, skeletal diameter, surface area, and perimeter. The aspect ratio (AR) was calculated by dividing the mitochondrial skeletal length by the mitochondrial skeletal diameter. By approximating mitochondria to elliptical structures, the mitochondrion circularity factor (4π × area/perimeter^2^) was used as a measure of mitochondrial elongation [107].

### 4.6. AEQ-Based [Ca^2+^] Measurements

To analyse local (mitochondrial, cytosolic and ER) cell Ca^2+^ levels, 24 h after TDP-43 silencing, cells were transiently transfected with the suitably targeted AEQ cDNA construct by means of the Ca^2+^-phosphate-based method [108]. 48 h after transfection, [Ca^2+^] measurements were performed using a computer-assisted luminometer equipped with a perfusion system, as previously described [103,104,105].

Briefly, for measuring [Ca^2+^]_mit_ and [Ca^2+^]_cyt_ transients, transfected HeLa cells were incubated in modified Krebs Ringer buffer (KRB: 125 mM NaCl, 5 mM KCl, 1 mM Na_3_PO_4_, 1 mM MgSO_4_, 5.5 mM glucose, 20 mM HEPES, pH 7.4, 37 °C) supplemented with 1 mM CaCl_2_ and the prosthetic group coelenterazine (5 µM, cat. no. sc-205904, Santa Cruz Biotechnology, St. Louis, MO, USA) to reconstitute functional AEQ_mit_ or AEQ_cyt_ (1.5 h, 37 °C). Then, coverslips were placed onto a recording chamber located into a thermostatted (37 °C) luminometer and cells were perfused with CaCl_2_ (1 mM)-containing KRB (CaCl_2_-KRB) for 20–30 s. Ca^2+^ transients were elicited by stimulating cells with 100 μM histamine in CaCl_2_-KRB.

For measuring [Ca^2+^]_ER_, AEQ_ER_ was reconstituted by incubating HeLa cells firstly in EGTA (1 mM)-containing KRB (10 min, 37 °C) and then in KRB supplemented with 5 µM ionomycin (Sigma-Aldrich), 500 µM EGTA and 5 µM coelenterazine-n (cat. no. 82260, AnaSpec, Fremont, CA, USA) (1 h, 4 °C). During these steps, ER Ca^2+^ stores are fully depleted. Coverslips were then transferred to the recording chamber and the experiment started by perfusing cells with KRB containing: 500 μM EGTA (2 min); 1 mM EGTA and 2% (*w*/*v*) BSA (3 min); 500 μM EGTA (2 min); 100 μM EGTA (1 min). Finally, cells were perfused with 1 mM CaCl_2_ (20 s) to accomplish ER Ca^2+^ replenishment. [Ca^2+^]_ER_ steady state was calculated by averaging the values of each experimental trace over 5 s after complete ER replenishment was achieved. Following these steps, cells were subsequently perfused with KRB containing 100 μM histamine to stimulate Ca^2+^ release from the ER through IP_3_-stimulated channels only in experiments in which the ER-Ca^2+^ discharge was evaluated. The release rate of Ca^2+^ from the ER was calculated by averaging the slope of the best-fit of each [Ca^2+^]_ER_ trace over the first 5 s after histamine addition.

At the end of each experiment, to discharge the remaining AEQ pool, cells were lysed using a hypotonic solution containing 10 mM CaCl_2_ and 100 μM digitonin. Recorded aequorin luminescence data were calibrated off-line into [Ca^2+^] values using a mathematical algorithm based on the [Ca^2+^] response curves of WT or mutated, low-affinity AEQs [109].

### 4.7. WB Analysis

HeLa cells, grown onto 12-wells plates and treated according to the different experimental conditions, were washed twice with ice-cold PBS and lysed in an ice-cold lysis buffer (60 µL/well) containing 10% (*w*/*v*) glycerol, 2% (*w*/*v*) SDS, 62.5 mM Tris/HCl, pH 6.8 and cocktails of protease and phosphatase inhibitors. Cell lysates were centrifuged (14,000× *g*, 10 min, 4 °C) to pellet cell debris, and the total protein content in the supernatant was determined by the bicinchoninic acid assay kit (Thermo Fisher Scientific) according to the manufacturer’s instructions. Samples were diluted to the desired concentration in the above buffer added with 50 mM dithiothreitol and 0.004% (*w*/*v*) bromophenol-blue, and boiled for at least 5 min. 10–20 µg of cell lysates were separated by SDS-PAGE (using 10% (*w*/*v*) acrylamide-N,N′-methylenebisacrylamide (37.5:1 (*w*/*w*)) or Mini-Protean TGX precast gels (4–20%, Bio-Rad Laboratories, Hercules, CA, USA)) and electro-blotted onto polyvinylidene difluoride (PVDF) membranes (0.45 μm pore size; Bio-Rad Laboratories). Membranes were stained with Coomassie brilliant blue solution (50% (*v*/*v*) methanol, 7% (*v*/*v*) acid acetic, 0.01% (*w*/*v*) Coomassie brilliant blue (Sigma-Aldrich) to verify even protein loading and transfer, and digitalised images were collected for subsequent densitometric analysis (see below). After destaining using pure methanol, membranes were washed three times with Tris-buffered saline (TBS, 20 mM Tris-HCl, pH 7.6, 150 mM NaCl) and incubated in a blocking solution made of TBS added with 0.1% (*w*/*v*) Tween-20 (TBS-T) and 3% (*w*/*v*) BSA (Sigma-Aldrich) (1 h, RT). Membranes were then incubated with the desired primary Ab diluted in TBS-T containing 1% (*w*/*v*) BSA (overnight, 4 °C). After three washes with TBS-T, membranes were incubated (1 h, RT) with horseradish peroxidase-conjugated anti-mouse, anti-rabbit (Sigma-Aldrich, cat. no. A9044 and A0545, respectively) or anti-goat (Santa Cruz Biotechnology, cat. no. sc-2354) IgG secondary Ab, depending on the primary Ab. Immunoreactive bands were visualised and digitalised by means of a digital camera workstation (NineAlliance, UVITEC, Eppendorf, Hamburg, Germany) using an enhanced chemiluminescence reagent kit (Millipore, Burlington, MA, USA). To perform densitometric analyses, the intensity of the immunoreactive bands was normalised to the optical density of the corresponding Coomassie blue-stained lane [110]. When analysing GSK3β phosphorylation, samples were run in parallel in the same gel and then probed with either the Ab to the phosphorylated Ser9 form (anti-p-GSK3β) or to the total protein (anti-GSK3β. To perform the densitometric analysis of p-GSK3β for each sample, the p-GSK3β immunoreactive band intensity was normalised to the corresponding band intensity of the total GSK3β signal.

The following primary Abs were used: anti TDP-43 mouse mAb (1:1000, cat. no. sc-376532 Santa Cruz Biotechnology, St. Louis, MO, USA); anti TOM20 rabbit pAb (1:1000, cat. no. sc-11415, Santa Cruz Biotechnology, St. Louis, MO, USA); pan anti IP3R rabbit pAb (1:500, cat n. ab97823, Abcam, Cambridge, UK), anti SERCA goat pAb (1:500; cat. no. sc-8094, Santa Cruz Biotechnology, St. Louis, MO, USA); anti GSK3β rabbit mAb (1:1000, cat. no. 9315, Cell Signaling Technology, Danvers, MA, USA); anti phospho-GSK3β (Ser9) rabbit mAb (pGSK3, 1:1000, cat. no. 5558, Cell Signaling Technology, Danvers, MA, USA); anti calnexin rabbit pAb (CLNX, 1.1000; cat. no. ADI-SPA-865, Stressgen, San Diego, CA, USA); anti calreticulin rabbit pAb (CRT, 1:500; cat. no.ADI-SPA-600-J, Stressgen, San Diego, CA, USA); anti MCU rabbit pAb (1:1000; cat. no. HPA016480, Sigma-Aldrich, St. Louis, MO, USA); anti MICU1 rabbit pAb (cat no. HPA037479, Sigma Aldrich, St. Louis, MO, USA); anti MICU2 rabbit pAb (HPA045511 Sigma Aldrich, St. Louis, MO, USA); anti MFN1 rabbit pAb (1:500, Santa Cruz Biotechnology, St. Louis, MO, USA ); anti MFN2 rabbit pAb (1:500, cat. no. ab56889, Abcam, Cambridge, UK); anti GRP75 mouse mAb (1:500, sc133137, Santa Cruz Biotechnology, St. Louis, MO, USA); anti GRP78 rabbit pAb (1:1000; cat. no.3183 Cell Signaling Technology Danvers, MA, USA), anti OXPHOS (1:1000, cat. no. ab110411, Abcam, Cambridge, UK).

### 4.8. RNA Isolation and Real-Time RT-PCR

RNA extraction, reverse transcription, and real-time PCR were conducted as previously described [111]. Briefly, 500 µL/well of tripleXtractor (GRISP Research Solutions, Porto, Portugal) was used to extract RNA from cell cultures according to the manufacturer’s protocol. Prior to RNA extraction, cell cultures were washed in PBS to remove the excess medium, then tripleXtractor was added directly on the dish and cells were detached using a cell scraper. RNA quantity and quality were tested by UV spectrophotometry and with electrophoresis by 2100 Agilent Bioanalyzer (Agilent Technologies, Santa Clara, CA, USA) following the protocol provided by the manufacturer. Then, 1 µg of RNA was retrotranscribed into cDNA using SuperScript II Reverse Transcriptase (Thermo Fisher Scientific), and gene expression was evaluated in a CFX 384 Real-Time polymerase chain reaction (PCR) System (BioRad) using the EvaGreen chemistry (Solis ByoDyne) and oligonucleotides reported in Appendix A. Thermocycler was set as follows: activation step (×1) 95 °C for 12 min; PCR Cycle (×40) 95 °C for 15 s (denaturation), 60 °C for 20 s (annealing), 72 °C for 35 s (elongation); final elongation (×1) 72 °C for 3 min; dissociation curve (×1). The original expression level was calculated as 2^{−[cycle threshold (C_t_, gene of interest)—C_t_ (housekeeping gene)]}. Tropomyosin 2 (TPM2) was used as the housekeeping (reference) gene. The mRNA amount for each target gene was normalised to the mRNA amount of the reference gene.

### 4.9. Statistical Analysis and Graphics

Statistics were based on non-parametric Mann–Whitney U test or Kruskal-Wallis H test, as indicated in the figure legends, with a *p*-value < 0.05 being considered statistically significant. ORIGIN (OriginLab corporation, Northampton, MA, USA) or Microsoft Excel (Microsoft Corporation, Redmond, WA, USA) software were used for statistics and graphics. In the box plots reported in figures, the upper and lower box range indicate the 1st and 3rd quartiles while lines, squares and whiskers in the boxes represent the median, the mean, and the 5 and 95 percentiles, respectively. *n*  =  number of at least three independent experiments or cells from at least three different transfections, as specified in the figure legends. In experiments of gene expression analysis, data are from at least four biological replicates and three technical replicates for each condition. In these cases, the mean value of the technical replicates for each biological replicate was firstly calculated, and data were then reported as the mean of the average biological replicate values.

## 5. Conclusions

TDP-43 depletion from the cell nucleus and its accumulation in cytosolic aggregates are major hallmarks of ALS. Consequently, TDP-43 may cause neurodegeneration by either gaining toxic potentials or losing its physiologic functions or both. Our study revealed a link between TDP-43 expression and MERCs formation/maintenance, whose deregulation has been previously related to ALS. We suggest that TDP-43 acts on MERCs by regulating both GSK3β expression levels and activation state. By considering that GSK3β is involved in many other fundamental functions, it is then possible to speculate that TDP-43 loss of function may cause other disease-related effects.

The present results demand further studies to obtain clearer insights into the mechanisms by which TDP-43 regulates GSK3β and MERCs.

## Figures and Tables

**Figure 1 ijms-22-11853-f001:**
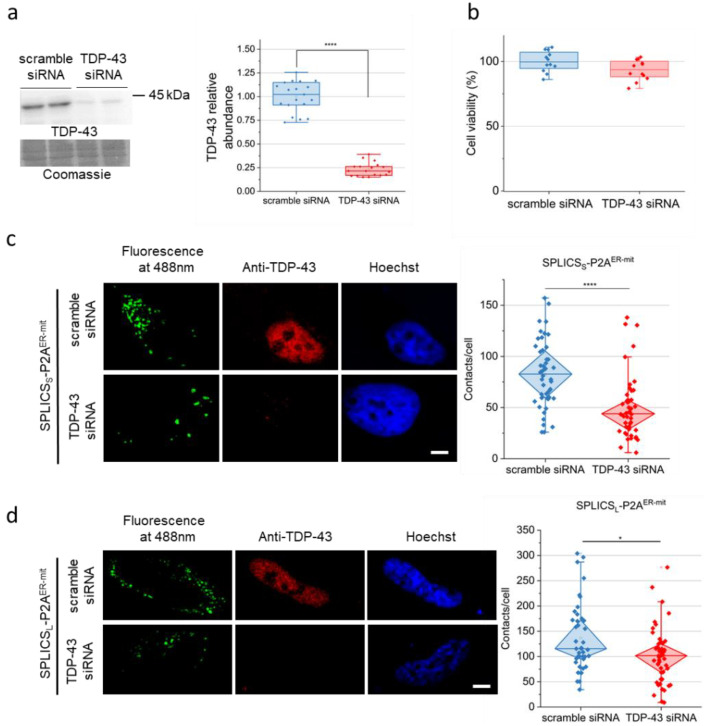
siRNA knock-down of TDP-43 does not influence cell viability but affects ER–mitochondria tethering in HeLa cells. (**a**) WB analysis of TDP-43 abundance in either TDP-43-silenced or control (scramble) HeLa cells. On the left, a representative WB and the corresponding Coomassie blue staining are shown. The box plot on the right reports the densitometric analysis of TDP-43 immunoreactive band intensity in the two cell populations. *n* = 18. (**b**) HeLa cell viability after scrambled or TDP-43 siRNA treatment, determined by the MTS assay. (**c**,**d**) MERCs density was evaluated by confocal microscopy on TDP-43-silenced and control HeLa cells using plasmids coding for either (SPLICS)^ER-mit^-short(S)-P2A (**panel c**) or for SPLICS^ER-mit^-long(L)-P2A (**panel d**) probes (green signal). Cells were also stained with an anti-TDP-43 antibody (red signal) and the nuclear marker Hoechst (blue signal). In panel c and d, the white scale bar correspond to 5 µm. In the right part of each panel, the diamond box plot reports the quantification of the indicated SPLICS_S/L_-P2A dots, as described in Materials and Methods. In (**c**), *n*= 84 cells for scrambled siRNA and 86 cells for TDP-43 siRNA; in (**d**) *n*= 43 cells for scrambled siRNA, and 57 cells for TDP-43 siRNA. n.s., not significant, * *p* < 0.05, **** *p* < 0.0001 (Mann–Whitney U test).

**Figure 2 ijms-22-11853-f002:**
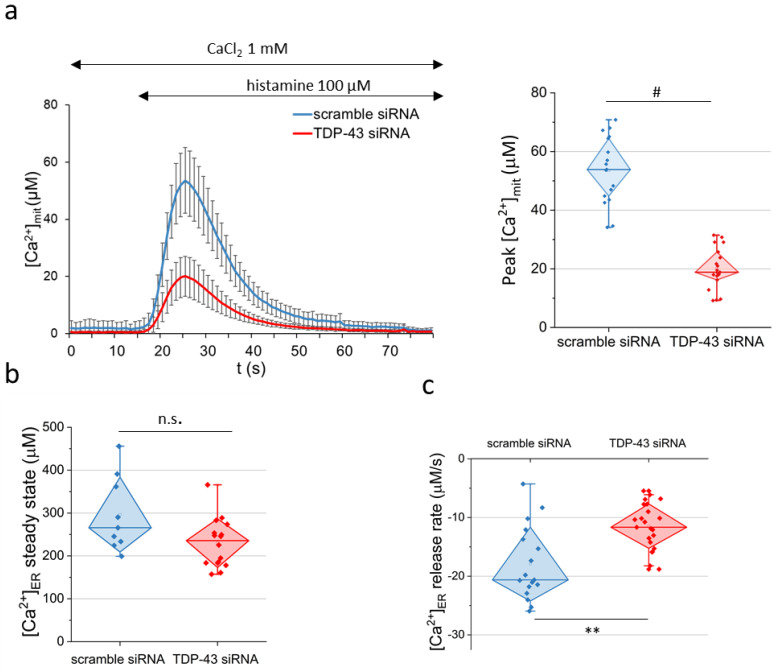
TDP-43 regulates both mitochondria Ca^2+^ uptake and ER Ca^2+^ discharge in HeLa cells. Ca^2+^ transients were measured in TDP-43-silenced or control (scramble) HeLa cells using AEQ probes targeted to the mitochondrial matrix (panel a) or the ER lumen (panels b and c). (**a**) Average kinetics of mitochondrial Ca^2+^ transients ([Ca^2+^]_mit_) upon stimulation with histamine (100 µM) at the indicated time point. Reported traces are mean ± SD (shown on the left). The box plot of the corresponding [Ca^2+^] peak value is shown on the right. *n* = 17, scrambled siRNA; *n* = 19, TDP-43 siRNA. (**b**) The box plot reports the mean value of the luminal [Ca^2+^]_ER_ steady state (see Materials and Methods). *n* = 9, scrambled siRNA; *n* = 16, TDP-43 siRNA; n.s., not significant (Mann–Whitney U test). (**c**) The box plot reports the luminal Ca^2+^ rate of release (expressed as µM/s) upon histamine stimulation (100 µM). *n* = 23, scrambled siRNA; *n* = 17, TDP-43 siRNA. ** *p* < 0.01; # *p* < 10^−7^ (Mann–Whitney U test).

**Figure 3 ijms-22-11853-f003:**
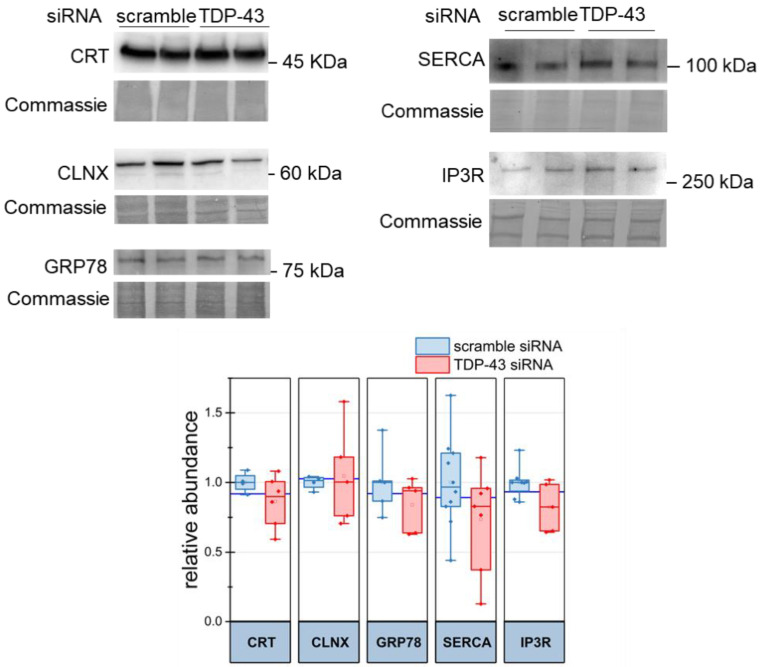
TDP-43 silencing does not modify the expression of proteins involved in ER Ca^2+^ homeostasis. The abundance of CRT, CLNX, GRP78, IP3R or SERCA was monitored by WB in both control (scramble) and TDP-43-silenced HeLa cells. The upper panel shows a representative WB for each analysed protein and the corresponding Coomassie blue staining of the membranes; the box plot in the lower panel reports the relative abundance (normalised to control samples) of each protein. *n* = 5, *p*-value > 0.999 (Mann–Whitney U test).

**Figure 4 ijms-22-11853-f004:**
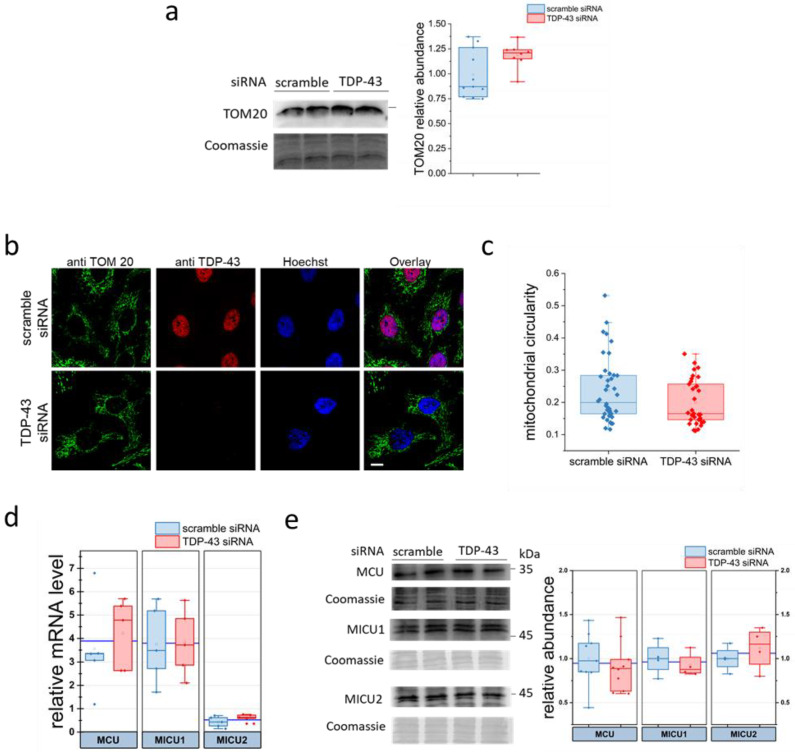
Mitochondria are unaffected by TDP-43 silencing in HeLa cells. (**a**) TOM20 expression was analysed by WB in lysates of HeLa cells silenced, or not (scramble), for TDP-43 expression. Both a representative WB (left panel) and the densitometric analysis for TOM20 expression (right panel) are reported; *n* = 8 for each cell type, *p*-value > 0.1. (**b**) Representative confocal images of TDP-43-silenced or control (scramble) HeLa cells stained with antibodies to TOM-20 and to TDP-43. Nuclei were visualised by Hoechst staining. Scale bar, 20 µm. (**c**) The box plot reports the values of mitochondrial circularity (see Materials and Methods) in both cell types. *n* = 41 cells for each condition, *p*-value > 0.06. (**d**,**e**) The relative abundance of MCU, MICU1 and MICU2 mRNAs (**d**) or proteins (**e**) were evaluated by RT-PCR or WB, respectively. In (**e**), representative WBs (left panel) and densitometric analyses of immunoreactive bands (right panel) are reported. *n* = 5, *p*-value > 0.5 (**d**); *n* = 4, *p*-value > 0.2 (**e**). Mann–Whitney U test for all statistics.

**Figure 5 ijms-22-11853-f005:**
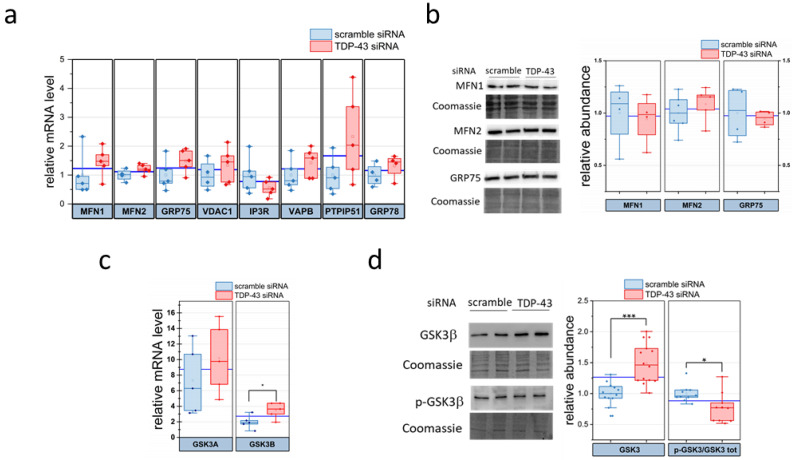
RT-PCR and WB analyses of proteins involved in ER–mitochondria tethering show increased expression and activation of glycogen synthase kinase-3β (GSK3β) inTDP-43-silenced HeLa cells. (**a**) RT-PCR analysis of targeted genes in TDP-43-silenced and control (scramble) HeLa cells. Due to high differences in relative mRNA levels between different genes, for the sake of clarity, in this diagram, data of each gene were further normalised to the corresponding control value. *n* = 4, *p*-value > 0.06. (**b**) HeLa cells lysates (with or without TDP-43 silencing) were analysed by WB for the expression of MFN1, MFN2 and GRP75. Representative WB images for the proteins of interest and the corresponding Coomassie blue-stained membrane are shown in the left panel. The right panel reports the densitometric analysis for the screened proteins. *n* = 4, *p*-value > 0.1. (**c**) RT-PCR analysis of GSK3A and GSK3B mRNA levels in TDP-43-silenced and control (scramble) HeLa cells. * *p*-value < 0.05, Mann–Whitney U test. (**d**) TDP-43-silenced and control (scramble) HeLa cells lysates were analysed by WB using antibodies against total GSK3β and phospho-Ser9 GSK3β (p-GSK3β). The left panel shows representative WB images, while the plots on the right show the abundance of total GSK3β and of the p-GSK3β/GSK3β ratio (see Materials and Methods). *n* = 14 for both scrambled and TDP-43 siRNA samples, for total GSK3β; *n* = 10 for p-GSK3β/GSK3β; * *p*-value <0.05, *** *p*-value < 0.01, Mann–Whitney U test for all statistics.

## Data Availability

Not applicable.

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
