# Peer review of "Regulation of Endoplasmic Reticulum–Mitochondria Tethering and Ca2+ Fluxes by TDP-43 via GSK3β"

_ijms, 2021, doi:10.3390/ijms222111853_

Round 1

Reviewer 1 Report

The manuscript perfectly fits the International journal of molecular sciences and will certainly attract attention of even broader audience, especially because it deals with very important topics in a very modern way. The objectives in the introduction are clearly formulated and the authors have collected an updated dataset using cutting edge methodology. The paper is really well written and structured, although there is issue that need to be solved right away such as in the abstract, in my opinion there is too much theory and too few research results and it should be modified.  

Given these mild shortcomings the manuscript requires minor revisions. I would like to congratulate on the quality of the work.

Reviewer 2 Report

The authors studied mitichondria-endoplasmic reticulum contacts (MERCs) and their Ca2+ fluxes with different genetic approaches in order to understand if the nuclear protein TDP-43's loss of function disrupts mitochondria-ER cross-talk and could be connected to neurodegenerative disorders such as ALS. With extensive experimental analysis they were able to establish that TDP-43 down-regulation decreases MERCs density, thus perturbing the Ca2+-based ER-mitochondria communication.  

This paper has a clear structure, the writing is easy to read and the cited references are numerous.

The only issue I find is with the extensive Figure cations. Although the Figures should be self-explanatory, the huge amounts of text below the images make it hard to read and understand. I therefore suggest that the authors reduce the text in Figure captions and move the more experimental details to the Materials and Methods section.

In general, the paper presents novel and interesting research work with relevance for neurodegenerative disorders like ALS. I recommend the publication of this paper in the Special Issue "Calcium Signaling in Human Health and Diseases 3.0" of the journal IJMS.
